# Penalty Decoding: Well Suppress the Self-Reinforcement Effect in Open-Ended Text Generation

**Wenhong Zhu , Hongkun Hao** and **Rui Wang**[*]
Shanghai Jiao Tong University
{zwhong714, haohongkun, wangrui12 }@sjtu.edu.cn

## Abstract

The decoding algorithm is critical for open-ended text generation, transforming latent representations into coherent and meaningful outputs. This paper investigates the self-reinforcement effect in text generation and the effectiveness of a repetition penalty to mitigate it. However, determining the optimal repetition penalty value is challenging. To tackle this, we propose a forgetting mechanism that disregards distant tokens, reducing the burden of penalty selection. In addition, we introduce a length penalty to address overly short sentences caused by excessive penalties. Our penalty decoding approach incorporating three strategies helps resolve issues with sampling methods deviating from factual information. Experimental results demonstrate the efficacy of our approach in generating high-quality sentences resembling human output.[1]

## 1 Introduction

Open-ended text generation tasks involve generating coherent and fluent output with limited input information (Holtzman et al., 2020). These tasks encompass various applications such as chit-chat dialog (Thoppilan et al., 2022), story generation (Mostafazadeh et al., 2016), and similar domains. Transformer-based models (Vaswani et al., 2017), which predict the probability of the next token during text decoding, have been widely adopted for such tasks. Among them, the GPT series, utilizing a decoder-only auto-regressive model, has demonstrated remarkable performance (Radford et al., 2019). The choice of decoding strategy plays a crucial role in determining the quality of text generation, not only in open-ended text generation but also in other natural language generation tasks.

Decoding strategies can be categorized into two types. One is deterministic methods, and another

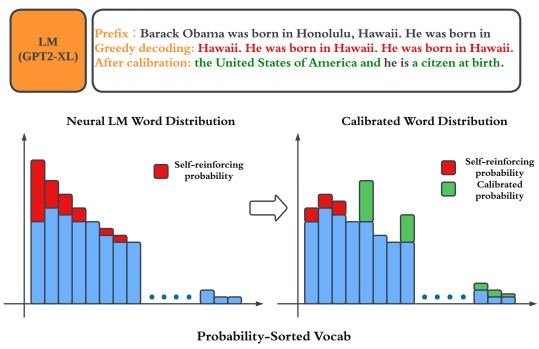

Figure 1: The predicted distribution of a neural language model (LM) can be regarded as a reinforced version of the model itself, as illustrated in the left part of the figure. To ensure high-quality text generation, penalties can be applied to the reinforced tokens, thereby correcting the distribution and improving the generated output.

is stochastic methods, also named truncation sampling (Meister et al., 2023). (1) Deterministic decoding strategies such as greedy search and beam search, which take the highest probability, would cause dull and repetitive text (Li et al., 2020). Contrastive search (Su et al., 2022), a recently proposed deterministic method, aims to enhance diversity while maintaining coherence in the generated text. However, it requires model retraining using contrastive training objectives and has a high time complexity of decoding. (2) Stochastic decoding strategies such as top-$k$ (Fan et al., 2018) and top-$p$ (Holtzman et al., 2020) sampling introduce randomness to increase the diversity of generated text. Several methods have been proposed to improve the truncation space based on these two methods, such as typical decoding (Meister et al., 2023), $\eta$-sampling (Hewitt et al., 2022), Mirostat (Basu et al., 2021) and other methods. However, it is worth noting that existing stochastic decoding strategies may exacerbate the issue of model hallucinations (OpenAI, 2023; Lan et al., 2022).

In this paper, we begin by reviewing and ana-

---

[*]Rui Wang is corresponding author.

[1]The source code and data will be shown at https://github.com/zwhong714/penalty_decoding.

lyzing the self-reinforcement effect proposed by Xu et al. (2022), which refers to the tendency of maximization-based decoding algorithms to assign higher probabilities to tokens that have already been generated, leading to repetitive text. Inspired by the work of Wang et al. (2022), who observed that the vanilla GPT3 model (Brown et al., 2020) often produces irrelevant and repetitive text, we hypothesize that this phenomenon may also exist in large language models. A critical insight of our work is to consider the distribution of a neural LM as an enhanced version. For instance, when presented with a prefix such as "Barack Obama was born in Honolulu, Hawaii. He was born in," the model (e.g., GPT2-XL) tends to repeat the previous context. This phenomenon might occur due to the probability distribution of the words "Hawaii" and "Honolulu" having a notable reinforcing effect within the probability space, as illustrated in the left part of Figure 1. To tackle this problem, we have introduced the repetition penalty proposed by Keskar et al. (2019). This penalty effectively mitigates the self-reinforcement effect and reshapes the token distribution, as demonstrated in the right section of Figure 1, resulting in a higher generation quality. However, tuning the repetition penalty hyperparameter can be challenging. Therefore, we propose the forgetting mechanism and length penalty as additional strategies to ensure generation quality. Through extensive experiments, we demonstrate the efficacy of our approach in generating sentences that closely resemble human-produced text.

## 2 Self-Reinforcement Effect

The self-reinforcement effect refers to the probability of repetition increasing with the number of historical repetitions (Holtzman et al., 2020; Xu et al., 2022). This effect can be intuitively reflected in the red probability shown in Figure 1, which may cause the model to get stuck in a repetition loop since the selection of tokens is forced to be limited to these enhanced tokens.

To quantify this phenomenon, we design the following metrics at various levels. Different from the metrics proposed by Xu et al. (2022), which analyze the self-reinforcement of repeated tokens by different numbers of repeated prefixes, we dynamically do analyzation in the model's natural decoding process to reasonably evaluate the self-reinforcement phenomenon arising from the natural decoding state of the model. Given a language

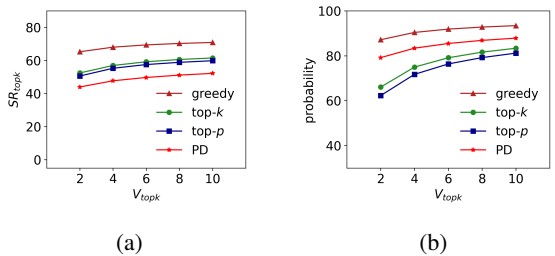

(a)             (b)

Figure 2: The comparison of different decoding methods in self-reinforcement effect. PD refers to our penalty decoding. a) The nucleus level ($SR_{topk}$) comparison. b) The summed probability of the top-$k$ tokens. **Note:** top-$k$ in $V_{topk}$ the size of token space with different values of $k$, while top-$k$ in the legend refers to the sampling method with $k = 5$.

model $\mathcal{P}_\theta$, we have:

**N-gram Level.** If $\frac{1}{n}\sum_{i=0}^{n-1}\mathcal{P}_\theta(x_{u+i}|x_{<u+i})$ is greater than $\frac{1}{n}\sum_{i=0}^{n-1}\mathcal{P}_\theta(x_{v+i}|x_{<v+i})$, where $x_u, ..., x_{u+n-1}$ is the next recurring $n$-gram of $x_v, ..., x_{v+n-1}$, $u$ and $v$ refer to the current decoding subscript position, then we say $n$-gram $x_u, ..., x_{u+n-1}$ is a reinforced version. We define the ratio of self-reinforcement at $n$-gram level $SR_n$ as:

$$SR_n = \frac{1}{L_g - n + 1} \#(\sum_{i=0}^{n-1} \mathcal{P}_\theta(x_{u+i} \mid x_{<u+i}) \\ > \sum_{i=0}^{n-1} \mathcal{P}_\theta(x_{v+i} \mid x_{<v+i})), \quad (1)$$

where $\#(\cdot)$ is the number of $n$-gram reinforced, $L_g$ is the length of the generated sentence.

**Nucleus Level.** The nucleus size of top-$k$ highest probabilities in the probability distribution for the $t$th token is defined as $NS(t) = \sum_{x \in V_{topk}} \mathcal{P}_\theta(x_t|x_{<t})$, where $V_{topk}$ is top-$k$ highest probability token space according to $\mathcal{P}_\theta$. If $\frac{1}{t}\sum_{u=1}^{t} NS(u) > \frac{1}{t-1}\sum_{u=1}^{t-1} NS(u)$, we say the model becomes much confident. Then

$$SR_{topk} = \frac{1}{L_g} \sum_{t=1}^{L_g} \mathbb{I}(\frac{1}{t}\sum_{u=1}^{t} NS(u) \\ > \frac{1}{t-1}\sum_{u=1}^{t-1} NS(u)), \quad (2)$$

where $\mathbb{I}$ is an indicator function; this metric measures how confident the model becomes.

| Method | $SR_1 \downarrow$ | $SR_2 \downarrow$ | $SR_3 \downarrow$ | $SR_4 \downarrow$ |
|---|---|---|---|---|
| greedy | 44.12 | 43.16 | 39.80 | 37.82 |
| top-$k$ | 23.71 | 10.83 | 6.88 | 4.81 |
| top-$p$ | 17.62 | 5.01 | 2.92 | 1.71 |
| penalty decoding | **5.36** | **2.71** | **1.32** | **0.84** |

Table 1: The self-reinforcement effect at $n$-gram level ($SR_n$) with different values of $n$. $p$ is 0.95 and $k$ is 5. Penalty decoding is with the parameter $\alpha = 1.5$ and window size $w = 100$.

**Results and Analyses.** We compare the self-reinforcement effect of three common decoding methods on a widely used dataset using the GPT2-XL model (see §4.1 for details). Table 1 illustrates this effect at different $n$-gram levels during text generation on the test set. It is observed that as the value of $n$ increases, the rate of reinforcement decreases. However, the self-reinforcement effect still remains significant in the case of greedy decoding for $n$-grams. This finding highlights the notable self-reinforcement result of greedy decoding, and the stochastic decoding algorithms can effectively mitigate this effect. Top-$p$ decoding demonstrates superior suppression, indicating a more pronounced mitigation as the sampling space expands.

Figure 2 presents the self-reinforcement effect at the nucleus level on the test set, along with the probability of the $V_{\text{topk}}$ space. As depicted in Figure 2(a), when employing greedy decoding the model tends to concentrate the probability more on the $V_{\text{topk}}$ space compared to the stochastic methods. Furthermore, Figure 2(b) demonstrates that the probability distribution heavily favors the $V_{\text{topk}}$ space after decoding, with the greedy search approach yielding probabilities as high as 90%. As is well known, as the generation proceeds, decoding to some extent concentrates on a smaller set of directions and greedy decoding would accelerate this process.

The above analysis surprisingly finds that adding perturbation to these reinforced spaces can effectively suppress this effect. However, stochastic methods depending on randomness would lead to serious hallucination (OpenAI, 2023; Lan et al., 2022). We guess other techniques exist to modify the maximization-based decoding, such as adding repetition penalties, to achieve the same result.

## 3 Method

In this section, we will present our penalty decoding approach, which comprises three techniques de-

signed to enhance the performance of greedy decoding by efficiently mitigating the self-reinforcement effect to generate high-quality text.

### 3.1 Repetition Penalty

The repetition penalty, as introduced by Keskar et al. (2019) (Keskar et al., 2019), aligns closely with our insight by penalizing tokens that could potentially exacerbate self-reinforcement. These penalties will be directly applied to the reinforced tokens. Following the softmax function, this reinforced portion can be redistributed to other tokens, as depicted in the green part of Figure 1, thereby ensuring the feasibility of alternative token sampling.

$$\mathcal{P}_\theta(v|\boldsymbol{x}) = \frac{\exp\left(v_i/\alpha \cdot (\mathbb{I}(i \in \boldsymbol{x}))\right)}{\sum_j \exp\left(v_j/\alpha \cdot (\mathbb{I}(j \in \boldsymbol{x}))\right)}, \quad (3)$$

where $v_i$ and $v_j$ represent specific tokens within the vocabulary, $\boldsymbol{x}$ represents the generated text, and $\alpha$ is a hyper-parameter greater than one.

### 3.2 Forgetting Mechanism

As illustrated in Equation 3, the penalties persist throughout the generation process and may result in significant semantic deviations. Thus, we propose the forgetting mechanism to limit repetition counting to a window $w$ around the current decoded location. This approach preserves text coherence by ensuring the decoding process aligns closely with contextual cues. The main implementation is as follows:

$$\mathcal{P}_\theta(v|\boldsymbol{x}) = \\ \frac{\exp\left(v_i/\alpha \cdot (\mathbb{I}(i \in \boldsymbol{x}[-w:]))\right)}{\sum_j \exp\left(v_j/\alpha \cdot (\mathbb{I}(j \in \boldsymbol{x}[-w:]))\right)}, \quad (4)$$

### 3.3 Length Penalty

Choosing an appropriate repetition penalty can be challenging, as discussed in Basu et al. (2021). If the repetition penalty is too small, it may not effectively alleviate self-reinforcement, while a large one can lead to short sentences as the <eos> [2] token is sampled early. The length penalty is applied to mitigate this problem and a straightforward implementation is a linear penalty imposed on the logits of <eos> token $\mathcal{P}_\theta(<eos>)$ as follows.

---

[2]<eos>: end-of-sentence identifier

$$\mathcal{P}_\theta(\text{<eos>}) = \alpha \cdot \mathcal{P}_\theta(\text{<eos>})(L_t - L_{\boldsymbol{x}}), \quad (5)$$

where $L_t$ is the preset target length, typically the same as the maximum length allowed, and $L_{\boldsymbol{x}}$ denotes the current length of the decoded text.

### 3.4 Penalty Decoding

We introduce the penalty decoding algorithm outlined in Algorithm 1 by incorporating the strategies above.

---

**Algorithm 1:** Penalty decoding

**Input:** Language Model $\mathcal{P}_\theta$; prefix $\boldsymbol{x}$;
 repetition penalty $\alpha$; window size $w$,
 targeted length $L_t$, the vocabulary of
 the language model $\mathcal{V}$.

**while** $\boldsymbol{x}[\text{-}1] \neq \text{<eos>}$ **do**

 Calculate the next token logits: $\mathcal{P}_\theta(v|\boldsymbol{x})$
 Let $w = \min(w, \text{len}(\boldsymbol{x}))$
 Let $\mathcal{P}_\theta(\text{<eos>}) =$
 $\alpha \cdot \mathcal{P}_\theta(\text{<eos>}) \cdot (L_t - \text{len}(\boldsymbol{x}))$
 Calculate Softmax function:

 $$\mathcal{P}_\theta(v|\boldsymbol{x}) = \frac{\exp\left(x_i/\alpha \cdot (\mathbb{I}(i \in \boldsymbol{x}[-w:]))\right)}{\sum_j \exp\left(x_j/\alpha \cdot (\mathbb{I}(j \in \boldsymbol{x}[-w:]))\right)}$$

 Get the most probable token $\hat{v}$:
 $\hat{v} = \arg\max_{v \in \mathcal{V}} \mathcal{P}_\theta(v|\boldsymbol{x})$
 Update the prefix $\boldsymbol{x} = [\boldsymbol{x} : \hat{v}]$

**end**

**Output:** The generated text $\boldsymbol{x}$.

---

## 4 Experiments

In this section, we examine the effectiveness of penalty decoding compared to other decoding methods for open-ended text generation tasks. Detailed setup for experiments can be found in Appendix A.1, and comprehensive ablation studies are shown in Appendix B.

### 4.1 Model and Dataset

In all of our experiments, we utilize the GPT2-XL (Radford et al., 2019) that is available in the Huggingface library (Wolf et al., 2020). The dataset is derived from WebText (Radford et al., 2019), specifically the held-out validation or test set of GPT-2.

### 4.2 Automatic Evaluation

We evaluate the quality of our generated texts using various automatic metrics, including diversity, MAUVE (Pillutla et al., 2021), coherence (Zhang et al., 2022), greedy ratio (Lan et al., 2022), and gen-length. Please refer to Appendix A.2 for more information about these metrics. Additionally, we measure the self-reinforcement effect after applying our penalty decoding.

**Results and Analysis.** The main results are presented in Table 2. Observations reveal that all stochastic decoding algorithms, except Mirostat, exhibit high diversity and MAUVE scores. This may be attributed to the incorporation of randomness, which diminishes the model's confidence and effectively mitigates the self-reinforcement effect. Furthermore, stochastic decoding is conducive to generating longer text.

Both greedy decoding and beam search are associated with reduced diversity and lower MAUVE scores. Greedy decoding, in particular, often generates longer texts, potentially due to the model becoming trapped in a repetition loop.

Our penalty-based decoding effectively strikes a balance among these automation metrics. In contrast to near-greedy decoding, it can generate longer sentences. Moreover, it achieves comparable levels of diversity and MAUVE scores, similar to contrastive decoding and other stochastic decoding methods. According to the greedy ratio, approximately 44.01% of tokens have been subjected to penalties in the generation, effectively mitigating the self-reinforcement effect at both the n-gram and nucleus levels, as demonstrated in Table 1 and Figure 2.

From Figure 2(b), we surprisingly find that our penalty decoding falls between greedy decoding and the stochastic decoding algorithms within the $V_{\text{topk}}$ space. This demonstrates our approach's capacity to mitigate model overconfidence and prevent hallucination problems caused by decoding divergence. In other words, as the generation process unfolds, penalty decoding still tends to focus on a narrower set of directions.

The coherence score demonstrates a strong correlation with the greedy ratio, where a higher greedy ratio often leads to enhanced coherence. Comparing the near-greedy approach that only utilizes repetition penalty, we observe penalty decoding can generate longer texts with higher MAUVE scores while maintaining the same level of coherence.

| Method | Diversity(%)↑ | MAUVE(%)↑ | Coherence↑ | Greedy Ratio(%) | Gen-Length |
|---|---|---|---|---|---|
| top-$k$ (Fan et al., 2018) | 85.67 | 92.19 | -1.74 | 59.91 | 93.08 |
| top-$p$ (Holtzman et al., 2020) | 91.79 | 94.50 | -2.13 | 53.56 | 91.99 |
| typical (Meister et al., 2023) | 93.44 | 93.77 | -2.26 | 52.02 | 91.57 |
| $\eta$-sampling (Hewitt et al., 2022) | 93.17 | 94.39 | -2.29 | 51.46 | 93.17 |
| Mirostat (Basu et al., 2021) | 53.01 | 69.26 | -1.27 | 87.49 | 96.48 |
| Greedy | 20.09 | 27.03 | -0.87 | 100.00 | 94.27 |
| Beam | 15.33 | 17.21 | **-0.68** | 92.21 | 85.24 |
| contrastive (Su et al., 2022) | 91.26 | 92.54 | -1.38 | 76.58 | 86.56 |
| near-greedy (Keskar et al., 2019) | **99.47** | 81.29 | -2.15 | 55.31 | 85.96 |
| penalty decoding | 98.73 | **95.43** | -2.15 | 55.99 | 94.49 |

Table 2: Test results of different decoding methods. The methods above the dotted line are stochastic decoding methods, while the others are deterministic. The hyper-parameter sweep is shown in Appendix A.3.

## 4.3 LLM for Evaluation

Conventional evaluation methods typically require human annotations and rely on ground-truth responses, which can be resource-intensive and time-consuming (Lin and Chen, 2023). We employ the large language model text-davinci-003 (Brown et al., 2020) to overcome these limitations as an evaluator surrogate. Details of the evaluation and prompt design can be found in the appendix C.

| Method A is better(%) | | Method B is better(%) | |
|---|---|---|---|
| penalty decoding | 44.33 | **55.67** | top-$k$ |
| | **55.44** | 44.56 | top-$p$ |
| | **54.69** | 45.31 | typical |
| | **54.45** | 45.55 | $\eta$-sampling |
| | 48.73 | **51.27** | contrastive |

Table 3: LLM evaluation results.

**Results.** The comparison results are displayed in Table 3, indicating that penalty decoding surpasses top-$p$, typical, and $\eta$-sampling decoding methods, and it offers decoding performance on par with the contrastive search method.

## 4.4 Human Evaluation

Auto evaluation metrics are not always entirely reliable. For instance, the MAUVE metric has demonstrated sensitivity to the length of generated content, as discussed by Su and Collier (2023). Therefore, we have conducted human evaluations to address this concern. Comprehensive details regarding the evaluation process can be found in Appendix D.

**Results.** The comparative results are displayed in Table 4. Human evaluations indicate that penalty decoding outperforms all other decoding methods.

## 4.5 Case Study

We present two examples to demonstrate the superior performance of penalty decoding compared

| Method A is better(%) | | Method B is better(%) | |
|---|---|---|---|
| penalty decoding | **66.00** | 34.00 | top-$k$ |
| | **76.00** | 24.00 | top-$p$ |
| | **64.00** | 26.00 | typical |
| | **72.00** | 28.00 | $\eta$-sampling |
| | **68.00** | 32.00 | contrastive |

Table 4: Human evaluation results.

to other decoding methods. The first tests whether the decoding method can generate factual information, while the second tests whether the decoding method can produce coherent text.

Table 7 demonstrates that our penalty-based decoding yields more factually accurate information about DeepMind company and produces sentences that closely resemble human-written text. While Contrastive decoding does manage to generate some factual content, the resulting text appears somewhat lacking in diversity. In contrast, the quality of stochastic sampling is unsatisfactory. In Table 8, the provided prefix offers sufficient contextual information, and our penalty decoding method successfully generates coherent text that maintains logical flow and coherence.

## 5 Conclusion

This paper delves into the self-reinforcement phenomenon in text generation and introduces penalty decoding as a solution. Through the integration of a repetition penalty, a forgetting mechanism, and a length penalty, the token distribution is adjusted to enhance diversity and diminish model-induced hallucinations, consequently elevating the quality and coherence of the generated text. Our penalty decoding can also be combined with sampling-based methods like top-$k$ and top-$p$ sampling. The findings and evaluations in this study aim to encourage further research in advancing the generation capabilities of neural language models.

## Acknowledgements

The authors are with the MT-Lab, Department of Computer Science and Engineering, School of Electronic Information and Electrical Engineering, and also with the MoE Key Lab of Artificial Intelligence, AI Institute, Shanghai Jiao Tong University, Shanghai 200204, China. This paper is supported by the General Program of National Natural Science Foundation of China (62176153), Shanghai Pujiang Program (21PJ1406800), Shanghai Municipal Science and Technology Major Project (2021SHZDZX0102), the Alibaba-AIR Program (22088682), and the Tencent AI Lab Fund RBFR2023012.

## Limitations

We acknowledge that our study has certain limitations. Firstly, we focus solely on an English open-ended text generation task and do not investigate the application of the proposed penalty decoding algorithm in other generation tasks. Additionally, we limit our experiments to relatively small models such as GPT2-XL, which consists of 1.7B parameters. The existence of the self-reinforcement effect in larger language models remains a conjecture and requires further experimental verification. Furthermore, the sample size we used for human evaluation is only 50, so the variance may be a bit large. We also do not explore the sensitivity of different models to repetition penalty or investigate the effects of alternative length punishment mechanisms, such as exponential punishment. Moreover, we do not conduct specific experiments to examine the performance of different decoding methods in generating factual information. This is an area that warrants further investigation. These limitations provide opportunities for future research.

## Ethics Statement

This paper will not pose any ethical problems. First, text generation is a standard task in natural language processing. Second, the datasets used in this paper have already been used in previous articles.

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

# A Experimental Details

## A.1 Expeimental Setups

We conducted all experiments using the GPT2-XL model. In the main experiment, we generated texts conditioned on the initial paragraph, limited to 32 tokens, from documents in the held-out set of WebText. The text generation process was terminated upon encountering an end-of-document token or reaching a maximum length of 128 new tokens. The test set consists of 1000 samples selected from Webtext, while the validation set comprises 500 samples extracted from the remaining data sets. The experimental configurations generally follow the work of Su and Collier (2023).

## A.2 Automatic Evaluation

**Diversity.** This metric considers the repetition of generated text at different $n$-gram levels and can be calculated as follows: **diversity** $= \prod_{n=2}^{4}(1.0 - \frac{\textbf{rep-n}}{100})$.

**MAUVE.** The score (Pillutla et al., 2021) is a metric that quantifies the similarity in token distribution between generated text and human-written text.

**Gen-Length.** This metric is utilized to compute the average length of the generated text.

**Coherence.** This metric (Su and Collier, 2023) employs the OPT-2.7B language model (Zhang et al., 2022) to assess the coherence between the generated text and a given prefix. The metric is defined as follows:

$$\frac{1}{|\hat{\boldsymbol{x}}|} \sum_{i=1}^{|\hat{\boldsymbol{x}}|} \log p_{\mathcal{M}}\left(\hat{\boldsymbol{x}}_i \mid [\boldsymbol{x} : \hat{\boldsymbol{x}}_{<i}]\right), \qquad (6)$$

where [:] is the concatenation operation.

**Greedy Ratio.** This metric quantifies the proportion of times the language model selects the token with the highest probability during text generation (Lan et al., 2022).

### A.3 Hyperparameters

Some decoding methods rely on specific hyperparameters that significantly impact the quality of the generated sentences. To determine the optimal hyperparameters, we utilize the MAUVE metric and search for a predefined set, as outlined in Table 5. The best-performing hyperparameters on the validation set, indicated in bold font, are then selected, and their corresponding performance on the test set is reported in Table 2.

| Method | Hyperparameters |
|---|---|
| top-$k$ | {3, 5, **7**, 9} |
| top-$p$ | {0.89, 0.90, 0.92, **0.95**, 0.99} |
| typical | {0.2, 0.9, 0.92, 0.95, **0.99**} |
| $\eta$ | {0.004, 0.002, 0.0009, **0.0006**, 0.0003} |
| Mirostat $\tau$ | {2, **3**, 4, 5} |
| Beam | {3, **5**, 7, 9} |
| contrastive | {**(5, 0.5)**, (5, 0.8), (10, 0.5), (10, 0.8)} |
| penalty | {1.1, **1.5**, 2.0, 2.5, 3.0} |

Table 5: Hyperparameter sweep for each decoding method. For contrastive, the parameter format is (k, penalty_alpha).

## B Ablation Studies

We employ the validation dataset for conducting ablation studies.

### B.1 The impact of repetition penalty

In the absence of the repetition penalty, the decoding performance is equivalent to that of greedy decoding. The diversity measure is only 20.09%, indicating limited variation in the generated text, while the MAUVE score is only 27.03%, indicating a lower resemblance to human-written text. As depicted in Figure 3(a), higher repetition penalties result in increased diversity but lower MAUVE

scores. Lower repetition penalties lead to decreased diversity and lower MAUVE scores.

### B.2 The impact of forgetting mechanism

A window size of 0 corresponds to greedy decoding, while a window size larger than the length of the generated text indicates near-greedy decoding. From Figure 3(b), we can observe that using a small window size results in lower text quality. Typically, a larger window value is essential to ensure the generation's quality, signifying that the self-reinforcement effect tends to persist throughout the building process. Nevertheless, it's worth noting that bigger window values are not always advantageous. In some cases, they can lead to a reduction in MAUVE, potentially due to the accumulation of penalties causing the text to deviate from its intended semantics, subsequently degrading text quality.

### B.3 The impact of length penalty

As shown in Table 6, the introduction of a length penalty has a discernible impact on the MAUVE metric for the generated text. It's crucial to recognize that MAUVE is influenced by text length and may not always guarantee the holistic quality of the generated content. However, it remains a pertinent observation that the accumulation of penalties can lead to the model generating overly short text.

## C Prompt Design

The design of prompts is illustrated in Figure 4. The Language Model (LLM) receives input consisting of a prefix, a reference completion, completion A generated by one decoding algorithm, and completion B generated by another decoding algorithm. The LLM then assesses and determines whether completion A is better than completion B based on criteria such as consistency, fluency, and informativeness. The evaluation process involves 200 samples, and any invalid outputs are excluded from the results.

## D Human Evaluation

We randomly selected 50 samples from the test set and applied various decoding algorithms to generate sentences from them. For evaluation, we engaged two English language experts. The evaluation method involves comparing sentences generated by penalty-based decoding with those produced by alternative decoding algorithms. Prior to

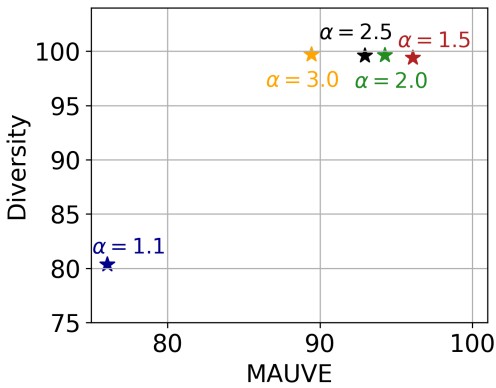

(a) The impact of repetition penalty

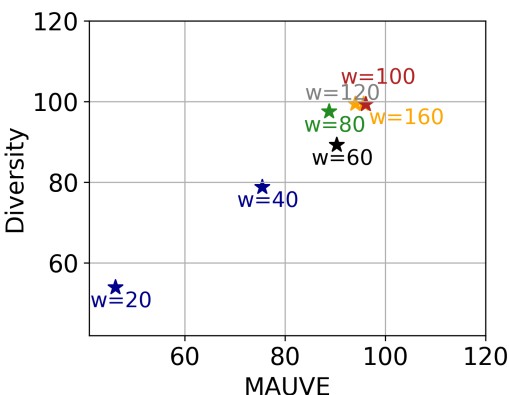

(b) The impact of forgetting mechanism

Figure 3: Ablation Studies

| Method | Diversity | MAUVE | Coherence | Greedy Ratio | Gen-length |
|---|---|---|---|---|---|
| **w** length penalty | 97.25 | **96.14** | -2.15 | 56.33 | **102.33** |
| **w/o** length penalty | **97.60** | 92.79 | -2.12 | 57.33 | 89.18 |

Table 6: The impact of length penalty

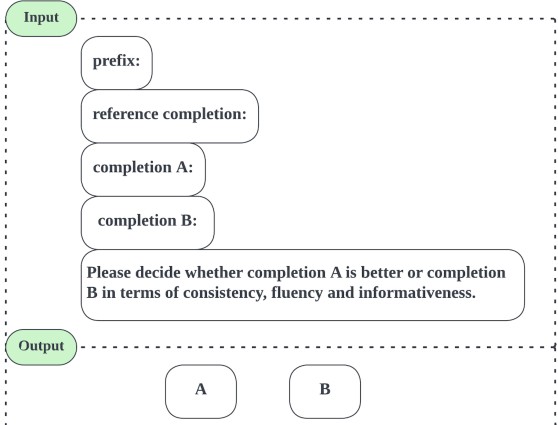

Figure 4: Prompt Design

the comparison, the experts are kept unaware of the specific decoding algorithm used for sentence generation. They should assess and select superior sentences based on three criteria: consistency, fluency, and informativeness.

| Prefix | DeepMind Company is |
|---|---|
| top-$k$ | the Google of artificial intelligence. The firm is responsible for developing DeepMind's flagship AlphaGo e program, and also has an interest in the field of deep learning. The company has also been involved in several other deep learning projects, such as a project to develop a neural network for image recognition. The company's main business is to help companies build and operate AI-enabled applications. The DeepMind team, |
| top-$p$ | —
In fact, Google has long had something of a habit of being named after companies.
The most recent and well-known example is Google's parent company Alphabet, which was named after the English scientist, Alexander von Humboldt.
Google parent company Alphabet is named after Alexander von Humboldt
Others include Google Fiber, which refers to the internet search company's fiber |
| typical | the company behind London DeepMind, a Google DeepMind project and a subsidiary of Alphabet (Google's parent company.) Google, in turn, is an investor in DeepMind, as well as the London-based artificial intelligence company SQM Technology Ltd. All the above investments (as well as many more), collectively, give DeepMind a valuation of $500 |
| $\eta$-sampling | co aware that they have received this notice. They are in discussion with us in accordance with the law and will respond accordingly.", the spokesperson wrote. The decision to suspend the project is an unusual one, because at the moment of the public announcement, it meant that the search for a successor wouldn't continue - for this search to continue, a firm would have to submit a formal bid.
Last |
| contrastive | a company founded by DeepMind co-founder Demis Hassabis and Alex Graves, who is also DeepMind co-founder. It is a research and development company that is building a deep learning system for artificial intelligence. The AI system has already won several computer vision and speech recognition comp |
| penalty decoding | a London-based artificial intelligence company that has been working on deep learning since 2010. It was founded by Demis Hassabis, a computer scientist and entrepreneur who cofounded Google's Deep Mind division in 2014 with Shane Legg at the University of Cambridge. The new AI system will be able to learn from data sets containing millions or billions of examples (called "training" datasets) as well as real world |

Table 7: Example 1. The generated documents by GPT-2-XL using different decoding methods with the prefix "DeepMind Company is". The factually correct content is highlighted in red.

| Prefix | In a shocking finding, scientist discovered a herd of unicorns living in a remote, previously unexplored valley, in the Andes Mountains. Even more surprising to the researchers was the fact that the unicorns spoke perfect English. |
|---|---|
| top-$k$ | "I was very excited and curious to find out more about these unicorns and their way of life," said Dr. John Catt, the lead author, who was studying the Andean clouded leopards in a remote location. 
 After a few weeks, he had his answer: The unicorns had a language of their own. 
 "They had very sophisticated social structures that were |
| top-$p$ | The wild creatures are a species of the same genus as the Asian hornless horse, which still continues to live in the wild. The researchers were able to record the unicorns' calls in a recording device, which recorded the animals' calls. 
 "Our study of the Andean valley Unicron - Latin for unicorn - is one of the most significant contributions to the history of speech in the world |
| typical | There are about 1,200 unicorn sightings, reported worldwide since the 1860s. Most of the sightings come from the southern hemisphere. Experts believe there could be up to 10,000 unicorns in total. 
 These creatures seem to be a different race of unicorns or possibly even a |
| $\eta$-sampling | After years of careful study of their behavior and language, researchers conclude that the unicorns live in these remote valleys in central Bolivia, and that there is a reason why we sometimes don't see them. Scientists have a name for such unicorns: the Andean llama herd, a wild species of goats native to the Andes that are endangered by loss of territory due to mining and grazing issues. Their |
| contrastive | According to the BBC, the researchers were able to record the unicorn's language in the region, where the animals live in the remote Andes Mountains. The unicorns, called the "Urania" in the study, are believed to be the largest herd of the animals in the world. 
 The research was carried out by researchers from Argentina's Universidad de La Plata, and the University |
| penalty decoding | The discovery is being hailed as one of the most important scientific discoveries ever made by an indigenous group from South America and has been dubbed "the unicorn language." The scientists believe it could be used for communication between humans and animals or even other species on Earth. 
 "We have found evidence that these creatures are capable of communicating with each other," said Dr. Carlos Bustamante, a researcher at Univers |

Table 8: Example 2. The generated documents by GPT-2-XL using different decoding methods with the prefix "In a shocking finding, scientist discovered a herd of unicorns living in a remote, previously unexplored valley, in the Andes Mountains. Even more surprising to the researchers was the fact that the unicorns spoke perfect English." The content that is relevant to the prefix is highlight in blue.