# OpenReview forum: "Penalty Decoding: Well Suppress the Self-Reinforcement Effect in Open-Ended Text Generation"
_EMNLP/2023/Conference — EMNLP 2023 Main_

### Official Review · Reviewer_FifM · 2023-07-20

**Typos Grammar Style And Presentation Improvements:** Equation 1 has a typo in the denomina…
**Soundness:** 3

**Excitement:**

3: Ambivalent: It has merits (e.g., it reports state-of-the-art results, the idea is nice), but there are key weaknesses (e.g., it describes incremental work), and it can significantly benefit from another round of revision. However, I won't object to accepting it if my co-reviewers champion it.

**Missing References:**

Contrastive Search Is What You Need For Neural Text Generation --- Su and Collier (TMLR 2023)

**Paper Topic And Main Contributions:**

This paper investigates the decoding strategy of text generation model and specifically focuses on the open-ended generation task. Starting from the self-reinforcement perspective, the author(s) propose two metrics, i.e., self-reinforcement rate on (i) N-gram and (ii) Nucleus level to demonstrate the self-reinforcement effect.

Based the above motivations, the author(s) design a simple penalty decoding method and compare it with other decoding approaches using a conventional LM and benchmark.

**Questions For The Authors:**

N/A

**Reasons To Accept:**

* The paper is well-written and easy-to-understand.
* Good analysis on the self-reinforcement effect and clever design of the evaluation metric.
* Comprehensive comparison with existing decoding method and the results are well-presented.
* Good limitation section clearly points out the limitations and future directions of this work.

**Reasons To Reject:**

* This work does not have human evaluation. While using LLM to evaluate the generations has been adopted by the community, the human evaluation results are indispensable to include especially for open-ended text generation.
* While the paper is well-presented, the technical novelty is a bit limited. That being said, it might be a good fit to a findings paper.
* [Inaccurate Reference] The coherence score is introduced by [1], it should be properly cited.

[1] Contrastive Search Is What You Need For Neural Text Generation --- Su and Collier (TMLR 2023)

**Reproducibility:**

5: Could easily reproduce the results.

**Reviewer Confidence:**

5: Positive that my evaluation is correct. I read the paper very carefully and I am very familiar with related work.

---

> ### Author Rebuttal · Authors · 2023-08-29
>
> We are genuinely grateful for your valuable comments.
>
> **Response for "Reasons To Reject"**
>
> - We randomly sampled 50 samples and performed manual evaluations in terms of coherence, fluency and informativeness. Our human evaluation results are as follows.
>
>     | Method A is better |                 | Method B is better                 |      |
>     | ------------------ | --------------- | ---------------------------------- | ---- |
>     | Penalty decoding   | $\textbf{66\%}$ | top-k (k = 7)                      | 34%  |
>     | Penalty decoding   | $\textbf{76\%}$ | top-p (p = 0.95)                   | 24%  |
>     | Penalty decoding   | $\textbf{64\%}$ | typical  (typical = 0.99)          | 36%  |
>     | Penalty decoding   | $\textbf{72\%}$ | $\eta$-sampling ($\eta$ = 0.0006)  | 28%  |
>     | Penalty decoding   | $\textbf{68\%}$ | contrastive （k = 5, $\alpha=0.5)$ | 32%  |
>
> Where A is our proposed penalty decoding algorithm.
>
> - We admit that the proposed method is simple, it is effective yet. Thank you for your comments.
>
> - We will add missing references and improve our writing. Thank you for your valuable suggestions.

---

### Official Review · Reviewer_tmfW · 2023-07-27

**Typos Grammar Style And Presentation Improvements:** See the reason to reject.
**Soundness:** 3

**Excitement:**

3: Ambivalent: It has merits (e.g., it reports state-of-the-art results, the idea is nice), but there are key weaknesses (e.g., it describes incremental work), and it can significantly benefit from another round of revision. However, I won't object to accepting it if my co-reviewers champion it.

**Missing References:**

N/A

**Paper Topic And Main Contributions:**

This paper aims at the self-reinforcement effect, which was identified by *Xu et al. (2022)*. To have a deeper probe of the problem, the authors design two new metrics on different levels to quantify the effect, i.e., N-gram level and Nucleus level. The measurement results show that maximization-based methods heavily suffer from the problem. So, this paper presents three strategies to alleviate the self-reinforcement effect in greedy search. The proposed strategies include repetition penalty, forgetting mechanism, and length penalty.

**Questions For The Authors:**

Question A. Have you measured the self-reinforcement effect after applying your penalty decoding?

Question B. Can the penalty decoding be used on sampling-based decoding methods, like top-k or top-p sampling?

Question C. Why do you introduce a limited window in the forgetting mechanism? In your words, it is designed for aligning factual information and contextual cues. I did not see any content about the factual verification in your evaluation part.

Question D. What is the major difference between your forgetting mechanism against the *penalized sampling* in *Keskar et al. (2019)*? It seems that the only difference is the newly introduced window. So, if this is the only difference, back to Question C.

**Reasons To Accept:**

1. Two new measurements regarding the self-reinforcement effect.

2. Three strategies aiming at alleviating the problems of repetition and short/long generated texts.

**Reasons To Reject:**

1. The writing and organization of this paper can be improved. This work primarily follows *Xu et al. (2022)* which proposed the self-reinforcement effect. So, I think there should be more comparisons between your work and this one, like what's your improvements over their work, and why you propose two new self-reinforcement metrics apart from their TP, IP, WR metrics. But I can hardly find the words or comparisons about *Xu et al. (2022)* in the paper.

I think this paper is a bit hard for readers from other fields to understand easily.

2. It is incremental work. The two kinds of new measurements of the self-reinforcement effect are rather interesting and nice. But the proposed method is trivial. The forgetting mechanism is highly similar to the *penalized sampling* proposed by *Keskar et al. (2019)*.

3. The experiments are not comprehensive and convincing. The improvements are marginal. On some metrics, the performance of the proposed method even is worse than the baselines. I cannot confirm the effectiveness of the method according to current experimental results.

**Reproducibility:**

4: Could mostly reproduce the results, but there may be some variation because of sample variance or minor variations in their interpretation of the protocol or method.

**Reviewer Confidence:**

4: Quite sure. I tried to check the important points carefully. It's unlikely, though conceivable, that I missed something that should affect my ratings.

---

> ### Author Rebuttal · Authors · 2023-08-29
>
> We are genuinely grateful for your valuable comments.
>
> **Response for "Reasons To Reject"**
>
> 1. We are sorry we didn't mention the differences in the paper. The experiment conducted by Xu et al. (2022) is to observe the self-reinforcement phenomenon of these repeated tokens by repeating prefix at different times. This may not reflect the problems that arise in the natural decoding state of the model. Therefore, this paper dynamically calculates these self-reinforcement metrics in the process of natural decoding of the model, so as to reasonably evaluate the self-reinforcement phenomenon caused by the maximum decoding algorithm of the model. Xu et al. (2022) mitigated this problem by proposing a new objective function, while we mitigated this problem by a new decoding algorithm in the inference phase. We will improve our writing and discuss more about the paper of Xu et al. (2022) in the next version of the paper.
>
> 2. As mentioned in this paper, Keskar et al. (2019) proposed repetition penalty. The forgetting mechanism is an improvement to the existing method, which is used to solve the problems of semantic deviation and token divergence of text decoding caused by simple punishment. Besides, our penalty decoding is a combination of three techniques including repetition penalty, forgetting mechanism, and length penalty. Besides, our experimental results show that the proposed penalty decoding method is effective.
>
> 3. We admit that we did not achieve SOTA on some indicators. Our newly proposed decoding algorithm has the same time complexity as greedy decoding, and has achieved comparable performance with other newly proposed decoding algorithms in some automatic evaluation indexes. Besides, we also do the human evaluation, which proves the effectiveness of the proposed method. Specifically, we randomly sampled 50 samples and performed manual evaluations in terms of coherence, fluency, and informativeness. Our human evaluation results are as follows.
>
>     | Method A is better |                 | Method B is better                 |      |
>     | ------------------ | --------------- | ---------------------------------- | ---- |
>     | Penalty decoding   | $\textbf{66}$% | top-k (k = 7)                      | 34%  |
>     | Penalty decoding   | $\textbf{76}$% | top-p (p = 0.95)                   | 24%  |
>     | Penalty decoding   | $\textbf{64}$% | typical  (typical = 0.99)          | 36%  |
>     | Penalty decoding   | $\textbf{72}$% | $\eta$-sampling ($\eta$ = 0.0006)  | 28%  |
>     | Penalty decoding   | $\textbf{68}$% | contrastive （k = 5, $\alpha=0.5)$ | 32%  |
>
>     Where A is our proposed penalty decoding algorithm.
>
>
> **Response for "Questions For The Authors":**
>
> **Answer A.** YES. The results are as follows.
>
> | Method                                           | $\textbf{SR}_1$ $\downarrow$ | $\textbf{SR}_2$ $\downarrow$ | $\textbf{SR}_3$ $\downarrow$ | $\textbf{SR}_4$ $\downarrow$ |
> | ------------------------------------------------ | ---------------- | ---------------- | ---------------- | ---------------- |
> | greedy                                           | 44.12            | 43.16            | 39.80            | 37.82            |
> | Top-k (k = 7)                                    | 23.75            | 10.92            | 7.07             | 5.02             |
> | top-p (p = 0.95)                                 | 17.35            | 4.96             | 2.97             | 1.85             |
> | Typical-sampling (typical = 0.99)                | 17.65            | 4.78             | 2.87             | 1.74             |
> | $\eta$-sampling ($\eta=0.0006$)                  | 18.07            | 4.87             | 2.75             | 1.58             |
> | penlty decoding ($\alpha=1.5, w=160$)  | 5.36  | 2.71 | 1.32 | 0.84 |
>
>
> **Answer B.** Yes. You can definitely use it in random sampling algorithms such as top-k, top-p, because these are just processing the token probability distribution. However, because these two algorithms themselves improve the diversity of the text, doing so may aggravate problems such as hallucination.
>
> **Answer C.** Yes, the purpose of setting up a window is to prevent the decoded tokens from being too scattered, that is, with a high degree of consistency with the previous text. As for how to design an indicator to evaluate the factual information of the text. We think it is a good research direction to detect whether the model produces illusions and make the output of the large model more convincing. In the appendix, we give two examples that show how well our decoding algorithm can produce factual information.
>
> **Answer D.** The difference is that we've added a forgetting mechanism and a length penalty. The former is used to ensure that the decoded text has a high degree of consistency with the previous prefix, and the length penalty is used to control the length of the sentence generation to avoid the problem of too short text caused by premature sampling of <eos> token.
>
>
>
> We will double-check typos and improve our writing in the next version of the paper. Thank you for your valuable suggestions.

---

### Official Review · Reviewer_rLC5 · 2023-08-03

**Typos Grammar Style And Presentation Improvements:** NA
**Soundness:** 4

**Excitement:**

3: Ambivalent: It has merits (e.g., it reports state-of-the-art results, the idea is nice), but there are key weaknesses (e.g., it describes incremental work), and it can significantly benefit from another round of revision. However, I won't object to accepting it if my co-reviewers champion it.

**Missing References:**

NA

**Paper Topic And Main Contributions:**

The paper focuses on developing an advanced decoding algorithm for open-ended text generation. It specifically investigates the self-reinforcement effect in text generation and the effectiveness of a repetition penalty to mitigate this effect.
The main contributions of this paper are two-fold:
1. Introducing new metrics to analyze the self-reinforcement effect.
2. Proposing penalty-based decoding methods that are designed to mitigate the self-reinforcement effect.
3. The proposed method performs quite well in automatic evaluation.

**Questions For The Authors:**

1. What are the differences between your setting of analyzing decoded text and the teacher-forcing setting? What's the intuition behind?
2. For now, the analyses remain at a shallow level.  With the generation proceeds, the decoding should to some extent concentrate on a smaller set of directions. How to divide this effect with your findings?

**Reasons To Accept:**

The key insight of this paper is quite inspiring. They disentangle the model probabilities into the model's true probability and reinforced probability. I really like this insight.
The authors propose new metrics for evaluating the self-reinforcement effect, which extends the teacher-forcing setting into a more general setting with decoded text.
The proposed methods perform well according to the automatic evaluation metrics.

**Reasons To Reject:**

- Major Concerns
1. Even though I really like the insight in the paper, I have to say the analyses and methods are not quite aligned with the insight. Specifically, the analyses mainly discuss the topic that different decoding methods incur different levels of self-reinforcement. The methods mainly penalize tokens that appear before within a certain window. I cannot see a concrete connection between them. You should improve the analyses part at least and discuss more in your methods.
2. Need human evaluation and maybe move ablation in main content:
Some recent papers find that automatic evaluation like mauve in open-ended generation is not that reliable. Also, the mauve metric is sensitive to length, so your length penalty maybe introduce some bias in your evaluation. You should add human evaluation to build more trust to your performance.
reference: https://arxiv.org/pdf/2210.14140.pdf

- Minor Concerns
1. The experiment setting is different from previous papers. For instance, in recent papers using webtext (e.g., truncation sampling), the test set is 5000 sentences and the completion length is 1024 tokens. This paper uses 1000 sentences and 128 completion. Can you please explain your settings?
References: https://arxiv.org/pdf/2210.15191.pdf
2. Notation is not clear enough. For example in definitions of Section 2. What are u and v, and what's the inputs? According to the following text, I guess the inputs are decoded outputs, where the setting is different from the teacher forcing setting used in original self-reinforcement paper. The authors should clarify and discuss more.

**Reproducibility:**

4: Could mostly reproduce the results, but there may be some variation because of sample variance or minor variations in their interpretation of the protocol or method.

**Reviewer Confidence:**

5: Positive that my evaluation is correct. I read the paper very carefully and I am very familiar with related work.

---

> ### Author Rebuttal · Authors · 2023-08-29
>
> We are genuinely grateful for your valuable comments.
>
> **Response for "Reasons To Reject"**
>
> 1. Thank you for affirming this article.
>
>     (a.) We apologize for the lack of clarity in the Self-reinforcement (analysis) section (section 2). In this section, we want to express that maximization-based decoding algorithms, such as greedy decoding algorithms, will lead to a serious self-reinforcement phenomenon. Random decoding algorithms can alleviate this phenomenon by introducing random noise, but often lead to hallucination. In view of this, we want to improve greedy search based on the insight proposed in this paper.
>
>     (b.) The proposed method is relatively simple yet effective, which is to punish tokens that have occurred before. However, this can cause the decoding to become divergent and semantic deviation occurs. In the section Appendix B, we do an analysis about the impact of window size. Figure 3(b) in the paper shows the result and proves the assumption aforementioned.
>
> 2. We randomly sampled 50 samples and performed manual evaluations in terms of coherence, fluency and informativeness. Our human evaluation results are as follows.
>
>     | Method A is better |                 | Method B is better                 |      |
>     | ------------------ | --------------- | ---------------------------------- | ---- |
>     | Penalty decoding   | $\textbf{66}$% | top-k (k = 7)                      | 34%  |
>     | Penalty decoding   | $\textbf{76}$% | top-p (p = 0.95)                   | 24%  |
>     | Penalty decoding   | $\textbf{64}$% | typical  (typical = 0.99)          | 36%  |
>     | Penalty decoding   | $\textbf{72}$% | $\eta$-sampling ($\eta$ = 0.0006)  | 28%  |
>     | Penalty decoding   | $\textbf{68}$% | contrastive （k = 5, $\alpha=0.5)$ | 32%  |
>
>     Where A is our proposed penalty decoding algorithm.
>
> 3. Our setting is generally following [A Contrastive Framework for Neural Text Generation](https://arxiv.org/abs/2202.06417)
>
> 4. We are sorry we didn't explain it clearly. $u$ and $v$ refer to the current decoding subscript position, respectively. Inputs are prefix plus currently decoded outputs. We will describe and change it in detail in the next version of the paper.
>
>
>
> **Response for "Questions For The Authors":**
>
> **Answer 1.** Language models are trained in the teacher-forcing manner, which is different from the auto-regressive decoding process during inference. Therefore, our setting of analyzing decoded text can reflect the real decoding process more accurately. The method set up in this paper is actually to achieve a corrective effect by suppressing the reinforcement part of each token and thus adjusting the overall probability distribution.
>
> **Answer 2.** Considering that simply imposing a penalty will cause the decoded tokens to constantly fall out of this small range, the purpose of adding a memory mechanism is essentially to control this range and ensure the diversity of decoding.

---

### Meta-Review · Area_Chair_WhdX · 2023-09-17

**Recommendation:** 4

**Metareview:**

Summary:

The paper investigates the self-reinforcement effect in open-ended text generation, and proposes two new metrics for measuring this effect. They also analyze the effectiveness of the repetition penalty to mitigate this effect, and propose a simple forgetting mechanism to improve the penalty selection process.

Strengths:

 1. Proposes two new metrics for measuring self-reinforcement effect in text generation.
 2. Propose a simple decoding strategy to mitigate the self-reinforcement effect to some extent.
 3. Experiments showing potential effectiveness of the proposed decoding approach.

Weaknesses:

1. Writing and organization of the paper could be improved for clarity, based on reviewers questions and concerns.
2. Additional in-depth analysis could help support its claims regarding the proposed forgetting mechanism.

---

### Decision · Program_Chairs · 2023-10-07

**Decision:**

Accept-Main

**Comment:**

Summary:

The paper investigates the self-reinforcement effect in open-ended text generation, and proposes two new metrics for measuring this effect. They also analyze the effectiveness of the repetition penalty to mitigate this effect, and propose a simple forgetting mechanism to improve the penalty selection process.

Strengths:

 1. Proposes two new metrics for measuring self-reinforcement effect in text generation.
 2. Propose a simple decoding strategy to mitigate the self-reinforcement effect to some extent.
 3. Experiments showing potential effectiveness of the proposed decoding approach.

Weaknesses:

1. Writing and organization of the paper could be improved for clarity, based on reviewers questions and concerns.
2. Additional in-depth analysis could help support its claims regarding the proposed forgetting mechanism.